# An In Vitro and In Vivo Translational Research Approach for the Assessment of Sensitization Capacity and Residual Allergenicity of an Extensive Whey Hydrolysate for Cow’s Milk-Allergic Infants

**DOI:** 10.3390/foods11142005

**Published:** 2022-07-07

**Authors:** Karen Knipping, Laura Buelens, Peter J. Simons, Johan Garssen

**Affiliations:** 1Danone Nutricia Research, 3584 CT Utrecht, The Netherlands; karen.knipping@danone.com (K.K.); laura.buelens@danone.com (L.B.); 2Utrecht Institute for Pharmaceutical Sciences, Faculty of Science, Utrecht University, 3584 CS Utrecht, The Netherlands; 3Polpharma Biologics B.V., 3584 CM Utrecht, The Netherlands; peter.simons@polpharmabiologics.com

**Keywords:** RBL, beta-lactoglobulin, whey hydrolysates, chimeric, monoclonal antibodies, IgE

## Abstract

Introduction: Hypoallergenic formulas prepared from hydrolyzed cow’s milk proteins are often used for the management of cow’s milk allergy (CMA) in infants. In this study, both in vitro assays and an in vivo mouse model for CMA were used to assess the sensitizing and allergenic potential of a newly developed, extensive whey hydrolysate (eWH). Methods: Gel permeation chromatography was used to characterize the molecular weight distribution of the peptides. Residual antigenicity was measured using a beta-lactoglobulin ELISA as well as with immunoblotting using anti-beta-lactoglobulin (BLG) and anti-alpha-lactalbumin antibodies. In vitro residual allergenicity was assessed using huFcεRIα-RBL-2H3 cells sensitized with anti-bovine BLG human IgE. In vivo sensitizing and allergenic potential was assessed in a CMA mouse model by measuring the acute allergic skin response, anaphylactic shock score, body temperature, serum mMCP-1, whey-specific IgE, and cytokines. Results: There was no in vitro residual antigenicity and allergenicity observed of the eWH. Mice sensitized with eWH showed no acute allergic skin reaction after challenge with whey, confirmed by an absence of whey-specific IgE and anaphylactic symptoms and decrease in body temperature and mMCP-1 levels. Conclusions: Results from our in vitro and in vivo translational approach to assess sensitization capacity and residual allergenicity indicate that the newly developed eWH is safe for use in CMA infants. This was subsequently confirmed in a clinical study in which this eWH was tolerated by more than 90% (with 95% confidence) of infants or children with confirmed CMA.

## 1. Introduction

Cow’s milk allergy (CMA) is a common allergy in the pediatric population with a prevalence of 2–3%, as shown in [1]. Food allergy is a condition in which the immune system reacts unusually to specific foods and can be IgE-mediated, occurring within seconds or minutes after exposure, or non-IgE-mediated, which can take much longer to develop, sometimes up to several days [2]. Food allergic reactions can occur in the skin, gastrointestinal tract, and respiratory tract or can even cause anaphylaxis. [3]. IgE-mediated reactions occur after re-exposure to an allergen. At first exposure of the allergen, a T helper 2 (Th2) response results in the production of allergen-specific IgEs, which will bind to the high-affinity IgE receptors (FcεRI) on mast cells or basophils (sensitization phase). When re-exposed to the allergen, cross-linking of the surface-bound IgE with the allergen will result in degranulation of the cells and release of mediators such as histamine, leukotrienes, and inflammatory cytokines (effector phase). Important factors for an effective degranulation is the concentration of allergen-specific IgEs bound to the cells, the amount of the allergen, and the IgE affinity for the allergen [4,5].

Whey proteins represent about 20% of the total bovine milk proteins. The major whey allergens are beta-lactoglobulin (BLG) and alpha-lactalbumin (ALA). When whey proteins are enzymatically degraded, peptides showing biological functions can occur, and some seem to have beneficial immunomodulatory properties [6,7,8,9]. Hypoallergenic infant formulas are usually made by enzymatically degrading cow’s milk proteins, sometimes followed by further modifications such as heat treatment or ultrafiltration, and are used for the management of CMA in infants. Formula characteristics can be assessed by biochemical analysis, and reduced allergenicity can be analyzed by using in vitro immunological assays, in vivo animal allergy models, and in vivo in humans using the skin prick test (SPT), patch test, or oral challenge tests [10].

New hypoallergenic compositions with improved treatment properties are needed, along with appropriate preclinical models, to evaluate these properties before introduction into humans. In the current study, the newly developed, extensive whey hydrolysate (eWH) was characterized on peptide level with molecular weight distribution (MWD) using gel permeation chromatography (GPC). Residual antigenicity was assessed using a BLG ELISA as well as gel electrophoresis, followed by immunoblotting with anti-BLG and anti-ALA antibodies. In vitro residual allergenicity was assessed by using RBL-2H3 cells, which express huFcεRIα and were sensitized with an oligoclonal pool of anti-BLG chuIgE antibodies. In vivo sensitizing capacity and allergenicity were investigated in a mouse model for CMA using female C3H/HeOuJ mice. After the in vitro and in vivo assessment of the eWH, a clinical study was conducted with the eWH, demonstrating that it did not evoke an allergic reaction in 90% (with 95% confidence) of infants or children with confirmed CMA [11]. With this translational strategy from in vitro to in vivo and eventually confirmation in a clinical study, we demonstrate the clinical relevance of the preclinical results, which will improve predictability of (residual) allergenicity of the newly developed cow’s milk hydrolysates.

## 2. Methods

### 2.1. BLG Protein ELISA

Residual BLG proteins were determined by performing the BLG ELISA (ELISA Systems, Windsor, QLD, Australia) according to manufacturer instructions. This BLG ELISA has been fully validated for the use of cow’s milk hydrolysates by Merieux NutriSciences with a LOQ of 0.2 ppm. In short, one gram of hydrolysate and controls were diluted in 10 mL extraction buffer and thoroughly mixed, and pH was adjusted (pH between 6.8–7.4). The samples were warmed in a water bath of 60 °C for 15 min, with shaking for one minute every five minutes. Standards and samples were added in duplicate to the pre-coated wells and, after 10 s shaking, incubated for 15 min at room temperature (RT). Plates were washed, and conjugate was applied to the wells and, after 10 s shaking, incubated for 15 min at RT. Plates were washed, and substrate solution was added and, after shaking for 10 s, incubated for 10 min at RT. The reaction was stopped using stop solution, and absorbance was read within 30 min using a plate reader (PowerWave HT; BioTek, Santa Clara, CA, USA) at 450 nm (corrected for the absorbance at 650 nm).

### 2.2. Sodium Dodecylsulfate-Polyacrylamide Gel Electrophoresis (SDS-PAGE) and Immunoblotting

The residual proteins pattern of the hydrolysate was assessed with SDS-PAGE (Bio-Rad Mini-PROTEAN III system) using a 4–20% TGX™ gel (Bio-Rad, Hercules, CA, USA). Protein standards (Bio-Rad) were used to estimate the molecular weight of proteins of interest. The hydrolysates were diluted 1:5 with reducing sample buffer (6.05 g Tris, 8.0 g SDS, 3.2 g dithiothreitol, 20 mg bromophenol blue in 60 mL H_2_O, and 40 mL glycerol 87%, pH 6.8), and 100 µg of protein was added to the gel. Proteins were either stained using silver staining or blotted to a PVDF membrane (Trans Blot Turbo Mini, Bio-Rad, Hercules, CA, USA).

Immunoblotting was done by incubating the PVDF membranes for 2 h in TBST with 2% gelatin followed by incubation with an anti-ALA and an anti-BLG monoclonal antibody (1:30,000; Bethyl Laboratories, Uden, The Netherlands) in TBST with 1% gelatin for 1 h. The binding of antibody was visualized by using Lumi-light Plus Western blotting substrate (Roche Diagnostics, Rotkreuz, Switzerland), and the chemiluminescence signal was measured with the Chemidoc XRS (Bio-Rad).

### 2.3. MWD of Peptides by Gel Permeation Chromatography (GPC)

The MWD of the peptides after hydrolysis was characterized using GPC. Hydrolysate samples were dissolved in 5 mg/mL HPLC-eluent, and undissolved particles were removed by centrifugation (5 min at 13,000× *g*) if required, followed by filtration (0.45 μm cellulose acetate filter). Size exclusion chromatography was performed with a Superdex Peptide 10/300 column (GE Healthcare) at 30 °C, with a flow rate of 0.9 mL/minute using an injection of 20 μL of a 5 mg/mL HPLC-eluent (7.4 g trifluoro acetic acid and 1173 g acetonitrile in 3500 g H_2_O) solution. The column was calibrated with 10 peptide standards: cytochrome C (M = 12,327), aprotinin (M = 6500), adrenocorticotropic hormone from porcine pituitary gland (M = 4567), insulin A-chain oxidized ammonium salt from bovine pancreas (M = 2532), angiotensinogen 1–14 renin substrate porcine (M = 1759), bradykinin salt (M = 1060), bradykinin fragment 1–7 (M = 757), bradykinin fragment 1–5 (M = 573), Ala-Ala-Ala-Ala-Ala (M = 373), and Gly-Leu (M = 188), all of which were from Sigma Aldrich, St. Louis, MO, USA. The eluate was monitored at 200 nm.

### 2.4. Degranulation of RBL-huFcεRI Cells (RBL-hεIa-2B12 Cells)

The cell-line RBL-hεIa-2B12, transfected with the α-chain of human (hu) FcεRI complex [12,13], was used for the RBL-huFcεRI degranulation assay. Degranulation of RBL-huFcεRI cells was performed as described previously [12]. Confluent growing RBL-huFcεRI cells (1 × 10^5^/well) in 96-well, flat-bottom culture plate were sensitized overnight with 3 μg/mL purified huIgE (Millipore) and stimulated with 10 μg/mL rabbit anti-huIgE antibodies (DakoCytomation) in Tyrode’s buffer for 1 h. huIgE/anti-huIgE served as a positive (pos) control (=100% degranulation). The sensitized cells with a pool of BLG-specific chimeric IgE monoclonal antibodies (Bioceros) were stimulated with anti-huIgE (10 µg/mL Tyrode’s buffer), BLG, or eWH (1 μg/mL in Tyrode’s buffer/HSA) for 1 h. The minimal degranulation (min) is the spontaneous degranulation of untreated cells and should always be below 20%. β-Hexosaminidase activity was measured by a fluorescence assay using 4-methylumbelliferyl-N-acetyl-α-D-glucosamine as a substrate [14]. The β-hexosaminidase released into the medium was calculated against the positive control (100% degranulation).

### 2.5. Animals

Three-week-old female C3H/HeOuJ mice (Charles River Laboratories, Erkrath, Germany) where kept for a minimum of 2 generations on a cow’s-milk-protein-free diet (Research Diet Services, Wijk bij Duurstede, The Netherlands) in which casein and whey are replaced by soy proteins to avoid tolerance to cow’s milk proteins. After arrival (CKP, Wageningen University, Wageningen, The Netherlands), mice were randomly allocated to one of the 5 groups (*n* = 8 per group): a negative control group (sensitized with PBS), a positive control group, an allergenicity assessment group with the newly developed eWH (both sensitized with whey), a sensitizing capacity group with the newly developed eWH, and a sensitizing capacity group with the control eHW. Mice were group housed (4/cage) in Makrolon type III cages per treatment, with 9 kGy irradiated sawdust bedding (Lignocel 9 s, J. Rettenmaier & Söhne GmbH, Rosenberg, Germany) and a mouse igloo with activation fast-trak (UNO) as environmental enrichment. Cow’s milk protein-free AIN93G [15] chow and tap water were available ad libitum. Mice were kept on light/dark cycle of 12 h/12 h, temperature 19–21 °C, and 42–47% relative humidity, and all procedures were performed during the light cycle. The animal study was registered under the code DAN202, and all experimental procedures were approved by an external, independent Animal Experimental Committee (DEC consult, Soest, The Netherlands) on 30 September 2011 and complied with national legislation and the principles of good laboratory animal care following the EU directive for the protection of animals used for scientific purposes.

### 2.6. Oral Sensitization

A schematic overview of the sensitization and challenge protocol is depicted in Figure 1. After two weeks of acclimatization, (5-week-old) mice were sensitized weekly (first sensitization is day 0) for a period of 5 weeks with 0.5 mL of a homogenized mixture of 33.2% WPC80 and 66.8% demineralized WPC, the extensive whey hydrolysate (eWH), or a previous used negative control extensive whey hydrolysate (nc eWH) [16] (40 mg/mL PBS) with cholera toxin (CT; 20 μg/mL PBS) as an adjuvant, per oral (p.o.) gavage (Gavage Feeding Needle (mouse 20–25 g), 20 G × 30 mm straight, 1.25 mm barrel tip (Fine Science Tools, Heidelberg, Germany). Control mice received CT and PBS. One week after the last sensitization, mice were sacrificed after oral challenge (100 mg/mL PBS) by cervical dislocation under isoflurane/O2 mixture anesthesia. Blood samples were collected by retro-orbital bleeding and centrifuged (15 min at 13,500 rpm) on the day prior to the first sensitization and on day 36. Sera were stored at −20 °C for measurement of whey-specific immunoglobulins, mouse mast cell protease-1 (mMCP-1), and cytokines.

### 2.7. Acute Allergen-Specific Skin Response

The acute allergen-specific skin response was measured after injection (30 G (yellow) needle (Terumo, Leuven, Belgium) for i.d. administration) of a mixture of 33.2% WPC80 and 66.8% demineralized whey protein concentrate or the eWH in the ear pinnae intradermally (i.d.). At day 36, mice sensitized orally with whey mixture, eWH, or nc eWH were injected i.d. in both ears with 20 μL whey mixture (1 mg/mL in PBS). On day 36, mice sensitized orally with WPC80 and demineralized whey protein concentrate mixture were injected i.d. in both ears with 20 μL eWH (1 mg/mL in PBS). Ear thickness was measured in duplicate using a digital micrometer (Mitutoyo, Veenendaal, The Netherlands). The allergen-specific net ear swelling was calculated by subtracting the basal thickness (0 h) from the thickness measured after 1, 4, and 24 h. The ear swelling is expressed as delta μm.

### 2.8. Shock Score

Degranulation of mast cells may cause an anaphylactic shock in severe allergic animals. To assess this anaphylactic shock severity, the animals were monitored closely, using a validated anaphylactic scoring (0, no symptoms; 1, scratching around nose and mouth; 2, swelling around the eyes and mouth, piloerection, reduced activity, higher breathing rate; 3, shortness of breath, blue rash around the mouth and tail, higher breathing rate; 4, no activity after stimulation, shivers, and muscle contractions [17]). Mice were placed on heating mats when a shock score of 3 was reached with a decrease in body temperature. The humane endpoint was set at an anaphylaxis score 4. Body temperature was measured with a temperature transponder (Plexx), which was placed on day 17 (Figure 1). The transponders were placed subcutaneously into the mice under isoflurane/O2 mixture anesthesia and were used to monitor the body temperature non-invasively at the timepoints t = 0, 15, 30, and 60 min.

### 2.9. Serum Mouse Mast Cell Protease-1 (mMCP-1)

Concentrations of serum mMCP-1 were analyzed with a mMCP-1 ELISA (mMCP-1 ELISA Ready-SET-Go!; eBiosciences, San Diego, CA, USA) according to manufacturers’ instructions.

### 2.10. Serum Whey-Specific Immunoglobulins

Levels of serum whey-specific IgE were analyzed by an in-house ELISA. Polystyrene high binding plates (Corning, Corning, NY, USA) were coated with 100 μL/well WPC80 (20 μg/mL) in coating buffer for 18 h at 4 °C. Plates were washed and blocked for 2 h with 0.5% HSA (Sigma Aldrich St. Louis, MO, USA). After washing, serum samples were applied in a 25× and a 75× dilution and incubated for 2 h at RT. Plates were washed and incubated with biotin-labelled rat anti-mouse IgE (BD Pharmingen, San Diego, CA, USA) at RT. Plates were washed and incubated with streptavidin-horseradish peroxidase 1:20,000 dilution (Sanquin, Amsterdam, The Netherlands) for 1 h. Plates were washed and developed with 1-Step Ultra TMB ELISA (Fisher Scientific, Waltham, MA, USA). The reaction was stopped with 4 M H_2_SO_4_ (Merck, Readington Township, NY, USA), and absorbance was measured at 450 nm on a plate reader (PowerWave HT; BioTek, Santa Clara, CA, USA).

### 2.11. Serum Th1/Th2 Cytokines

Serum cytokine levels of IL-2, IL-4, IL-5, IL-10 IL-12, IFN-γ, TNF-α, and GM-CSF were measured using a commercial Mouse cytokine Th1/Th2 magnetic bead immunoassay (BioRad, Hercules, CA, USA) on the Luminex Workstation Bio-Plex 200 System (BioRad, Hercules, CA, USA). The measurement was preformed according to the manufacturer’s instructions.

### 2.12. Statistical Analysis

The sample size calculation was based on the ear thickness as main parameter, using historical data, with a minimal effect size d = 0.25 and equal variation (0.17) for each experimental group. The mouse model data were analyzed using one-way ANOVA, followed by a Bonferroni’s multiple comparison post hoc test for selected comparisons. Data are represented as mean ± SD or as individual value plots. Statistical analyses were conducted using GraphPad Prism software version 8.0.0. Values of *p* < 0.05 were considered significant.

## 3. Results

### 3.1. MWD, Residual Proteins, and Antigenicity of Whey Hydrolysates

The MWD of the peptides of both eWH and nc eWH were characterized by GPC, showing a difference in % specifically in the ranges 1000–5000 Da and <500 Da. nc eWH showed a much larger proportion of peptides in the 1000–5000 Da range (appr 42%), whereas almost 60% of the peptides in eWH were below 500 Da (Table 1).

The BLG ELISA (limit of detection 0.2 ppm) showed no residual BLG proteins in eWH (0.06 ppm) and nc eWH (0.09 ppm), and both were well below the safety limit of 0.2 ppm, as is established for this specific BLG ELISA (Table 1).

Residual proteins in whey hydrolysates were determined with SDS-PAGE, and the antigenicity (capacity to bind to antibodies) was assessed with immunoblotting (Figure 2). Immunoblotting showed no binding of anti-BLG and anti-ALA antibodies to residual whey proteins in eWH and nc eWH.

### 3.2. In Vitro Residual Allergenicity

In vitro residual allergenicity was assessed by using RBL-2H3 cells expressing huFcεRIα, which were sensitized with a pool of anti-BLG chimeric humanized (chu)-IgE antibodies that cover the allergenic epitopes of BLG (Figure 3). The sensitized cells stimulated with anti-huIgE were set as 100% (pos). Incubation with intact BLG resulted in a degranulation of 85%, whereas incubation with both eWH and nc eWH showed no degranulation and were comparable to the background (min).

### 3.3. Shock Score and Body Temperature

In this study, there was only one drop-out in the positive control (whey/whey) group (*n* = 7). The anaphylactic shock score was measured 30 min after challenge as the most important read-out (Figure 4a). Four out of eight mice sensitized and challenged with a mixture of WPC80 and demineralized whey protein concentrate showed mild to severe anaphylactic shock symptoms, but none of the mice exceeded a shock score of 3. However, no shock symptoms were observed in the group sensitized with whey and challenged with eWH or in the whey-sensitized mice challenged with hydrolysates, indicating no allergic anaphylactic shock. Anaphylactic shock can cause a decrease in body temperature. The body temperature before allergen challenge of all animals was above 38 °C. A decrease to 37.3 °C was observed 30 min after challenge in mice sensitized and challenged with a mixture of WPC80 and demineralized whey protein concentrate (Figure 4b). All the mice in the other groups were able to maintain their body temperature, indicating that, as confirmed by an absence of an anaphylactic shock score, no allergic anaphylactic shock was observed in mice sensitized with whey and challenged with eWH or in the whey-sensitized mice challenged with hydrolysates.

### 3.4. Acute Allergen-Specific Skin Response

The acute allergen-specific skin response was measured after injection of whey protein or eWH in the ear pinnae intradermally (i.d.). The positive control (mice sensitized and challenged with a mixture of whey proteins) showed a significantly increased acute allergic skin response at 1 h compared to non-sensitized mice (Figure 5). Mice sensitized with the whey proteins mixture and challenged with eWH showed a reduction (non-significant) in ear swelling compared to the positive control, which was still significantly higher than non-sensitized mice. Mice sensitized with either eWH or nc eWH showed no acute allergic skin reaction (not significantly increased compared to non-sensitized mice) when challenged with whey protein, indicating that both hydrolysates were not able to sensitize to whey proteins.

### 3.5. Serum Mouse Mast Cell Protease-1 (mMCP-1), Whey-Specific IgE, and Th1/Th2 Cytokines

The concentrations of mMCP-1 in blood is widely used as a specific marker of mast cell activation after an allergic response. Mice sensitized and challenged with a mixture of whey proteins showed high levels of serum mMCP-1 (Figure 6). In contrast, mMCP-1 concentrations in the whey-sensitized mice challenged with eWH or in the mice sensitized with eWH and nc eWH and challenged with whey were not significantly different from non-sensitized mice, indicating that no mast cell degranulation occurred.

Food allergic symptoms are mostly IgE-mediated. Therefore, the measurement of allergen-specific IgE is an additional parameter to study sensitization. As expected, the allergen-specific IgE levels were strongly elevated in mice sensitized with a mixture of whey proteins, whereas sensitization with the hydrolysates did not induce a whey-specific IgE response (Figure 7).

The pattern and concentrations of cytokines secreted by Th1 and Th2 cells correlate with specific immune responses. The Th1 (IL-2, IL-10, IL-12, GM-CSF, IFN-gamma, and TNF-alpha)/Th2 (IL-4, IL-5, and IL-13) cytokine balance in serum was measured using a multiplex assay. The concentrations of IL-2 and IL-4 were below the detection limit of the assay. A reduction was found in serum cytokine concentrations IL-5 and IFN-gamma between the positive control group and the nc eWH (Figure 8). None of the other cytokines showed any differences between groups.

## 4. Discussion

The European Union introduced a regulation in 2007 on nutrition and health claims on foods to assure these claims are clear, accurate, and based on scientific evidence. For hydrolyzed formulas, it is key to affirm the absence of potential allergic epitopes. To assess the allergenic and sensitizing capacity of extensive whey hydrolysates, both in vitro methods and an in vivo mouse model of CMA were used in the current study.

After hydrolysis, the MWD of the peptides after hydrolysis showed that there were only peptides present <5000 Da, and SDS-PAGE showed no protein bands. There was no in vitro antigenicity (capacity to bind to antibodies) found by immunoblotting with anti-BLG/anti-ALA antibodies and in the BLG ELISA, indicating very low immunogenic properties of the eWH. This was further substantiated by the absence of in vitro allergenicity of the eWH in the RBL-huFcεRI degranulation assay, which has proven human relevance since the pool of chimeric antibodies cover the allergenic epitopes as assessed with epitope mapping with IgE of CMA infants and known from literature [12]. It should be noted that the other allergen in whey, ALA, is only assessed with the immunoblotting when using anti-ALA antibodies, while both the ELISA and RBL-huFcεRI are specifically investigating BLG. In a previous study, however, we established that during the hydrolysis, BLG and ALA are broken down at the same rate [18]

For the assessment of the hypoallergenic potential of the eWH in the CMA mouse model, mice were sensitized with whey and challenged with eWH. The positive control (i.e., both sensitization and challenge with whey) showed high levels of whey-specific IgE exemplified by a presence of allergic symptoms, acute allergic skin reactions, anaphylactic shock, decrease in body temperature, and elevated mMCP-1 concentrations. Mice sensitized with whey and challenged with eWH showed elevated levels of whey-specific IgE as expected. A reduction (non-significant) in ear swelling was observed compared to the positive control, but the ear swelling was significantly higher than found in non-sensitized mice, indicating a residual immune reaction to the eWH but not contributing to the allergic reaction since there were no anaphylactic symptoms and no decrease in body temperature or mMCP-1 observed with the eWH. Based on the overall assessment, the eWH is considered non-allergenic and thus safe for humans, including allergic individuals.

To assess the sensitizing potential of whey hydrolysates, mice were sensitized with either eWH or nc eWH and challenged with whey. Mice sensitized with either eWH or nc eWH showed no acute allergic skin reactions after challenge with whey, indicating that both hydrolysates were not able to sensitize to whey protein allergy. This was confirmed by the absence of whey-specific IgE and anaphylactic symptoms and decrease in body temperature and mMCP-1 after challenge with whey.

Allergic reactions are characterized by the activation of allergen-specific CD4+ T cells, which produces a specific set of cytokines, namely IL-4, IL-5, and IL-13. These cytokines trigger IgE class switching and the recruitment and activation of eosinophils and mast cells. IL-5 acts as an eosinophil stimulating factor [19]. Only the nc eWH showed a significant decrease of IL-5, which is in line with the absence of an allergic response, whereas the eWH shows lower concentrations of IL-5, which were, however, not significant. Th1-type cytokines (IL-2, IL-10, IL-12, GM-CSF, IFN-gamma, TNF-alpha) initiate cell-mediated immune responses, which are necessary for the defense against viruses and bacteria. IFN-gamma is a cytokine that is critical for regulation of immune responses against viruses and bacteria and bridges the innate and specific immune response pathways, but it also inhibits Th2 cell differentiation. Decreased levels of IL-12 and IFN-g have been found in asthma, and this might diminish their ability to inhibit IgE synthesis [20]. Even though the allergic response in the nc eWH was absent, together with decreased IL-5, the concentration of IFN-gamma is also significantly lower, which is an unexpected finding. The significance of these cytokines should be further explored in a time-course manner in the CMA mouse model.

Overall, using our in vitro and in vivo translational research approach for the assessment of sensitization capacity and residual allergenicity, our newly developed eWH can be considered safe for the use in CMA infants. In a subsequent clinical study that was recently published [11], this was demonstrated as well: the eWH does not evoke an allergic reaction in 90% (with 95% confidence) of infants with confirmed CMA and can thus be regarded as safe.

## Figures and Tables

**Figure 1 foods-11-02005-f001:**
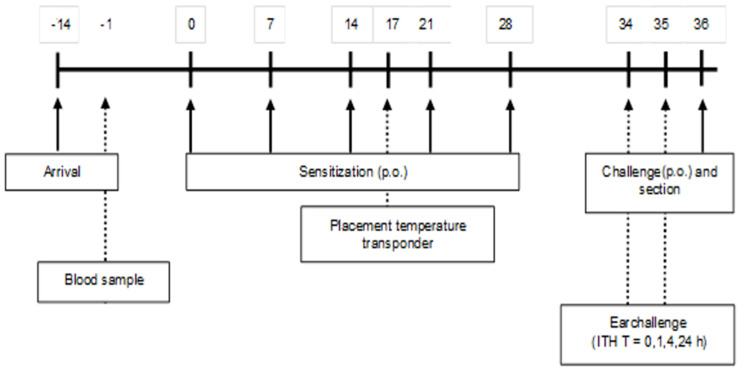
A schematic overview of the sensitization and challenge protocol. Five-week-old female C3H/HeOuJ mice (8 mice/group, 5 groups) were sensitized p.o. with 0.5 mL homogenized whey or whey hydrolysate (40 mg/mL PBS) with CT (20 μg/mL PBS) as an adjuvant. Control mice received CT plus PBS. Mice were boosted weekly for a period of 5 weeks. Blood samples were collected on days −1 and 36, and mice were sacrificed after p.o. challenge on day 36.

**Figure 2 foods-11-02005-f002:**
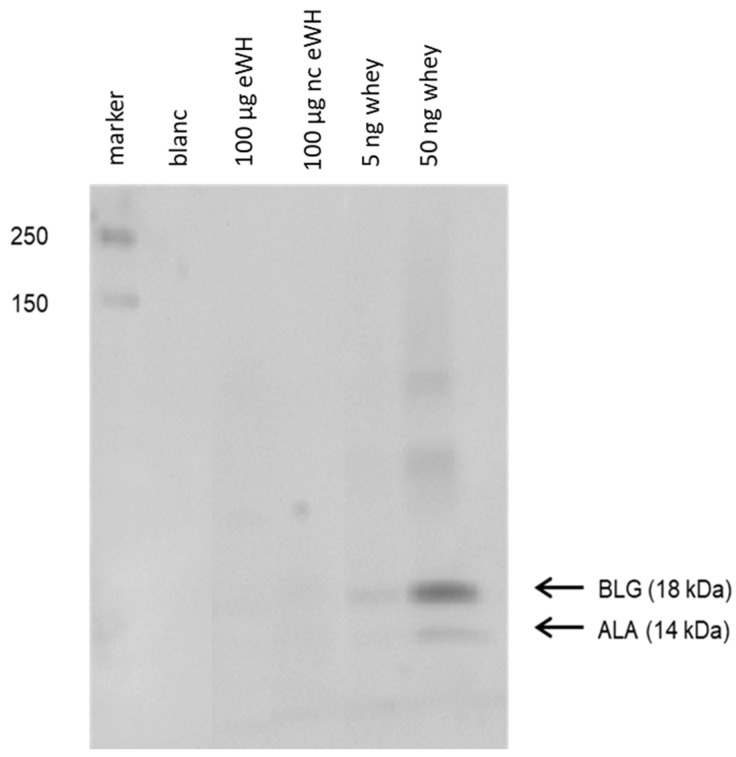
Immunoblotting with anti-BLG and anti-ALA antibodies. Western blotting of extensive whey hydrolysate (eWH), negative control extensive whey hydrolysate (nc eWH), and intact whey followed by incubation with anti-BLG and anti-ALA antibodies.

**Figure 3 foods-11-02005-f003:**
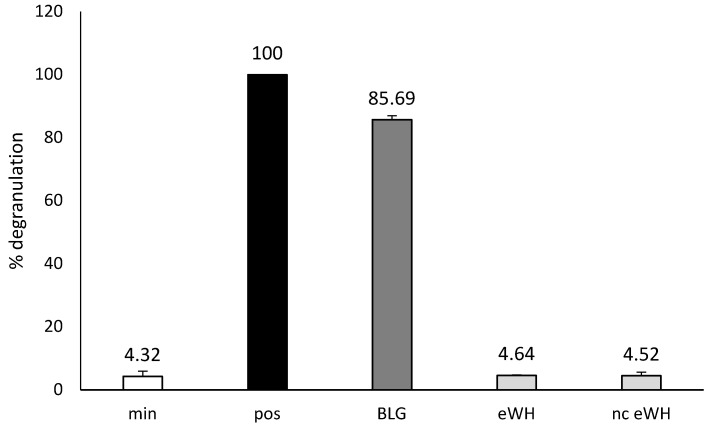
In vitro residual allergenicity by using RBL-2H3 cells expressing huFcεRIα after sensitization with an oligoclonal pool of anti-BLG chu-IgE antibodies. Degranulation with anti-human IgE was set to 100% (pos) and background degranulation is shown as “min”. Mean ± SD (*n* = 3) are shown.

**Figure 4 foods-11-02005-f004:**
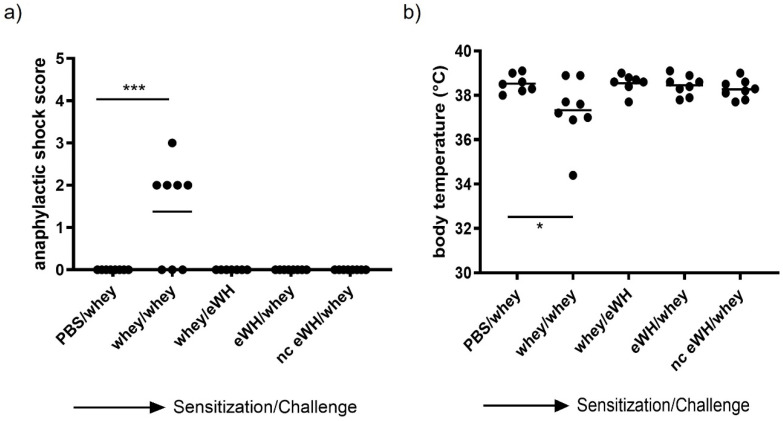
Anaphylactic shock score (**a**) and body temperature (**b**) 30 min after challenge. Anaphylactic symptoms and the body temperature were determined 30 min after i.d. whey challenge (day 36) in mice sensitized to whey or whey hydrolysates. Data are expressed as individual values. ANOVA for treatment was significantly different; post hoc Bonferroni tests showed significance *** *p* < 0.001, * *p* < 0.05 compared to non-sensitized mice (PBS/whey).

**Figure 5 foods-11-02005-f005:**
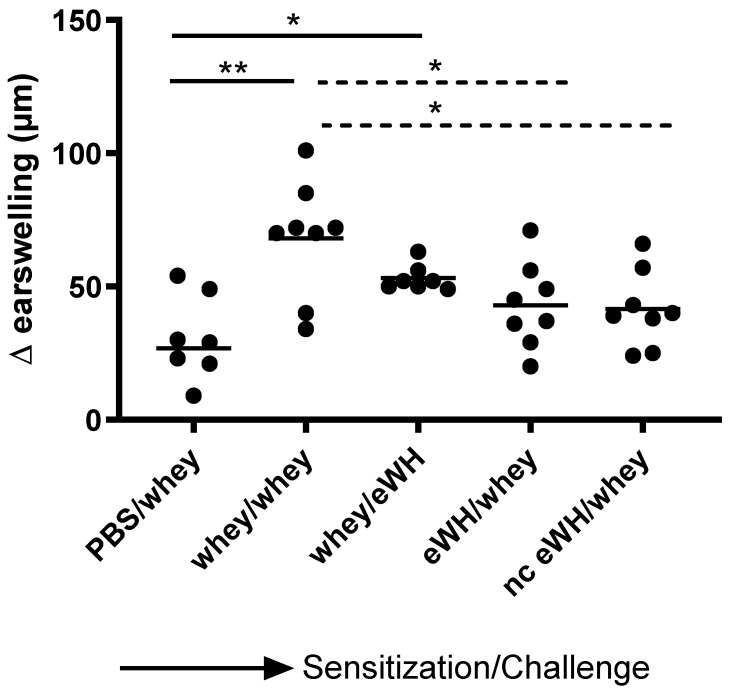
Acute allergen-specific skin response 1 h after challenge. The acute skin allergic response was measured 1 h after i.d. challenge on day 36 with whey or whey hydrolysate in both ears. Data are expressed as individual values. ANOVA for treatment was significantly different; post hoc Bonferroni’s multiple testing showed significance * *p* < 0.05, ** *p* < 0.01.

**Figure 6 foods-11-02005-f006:**
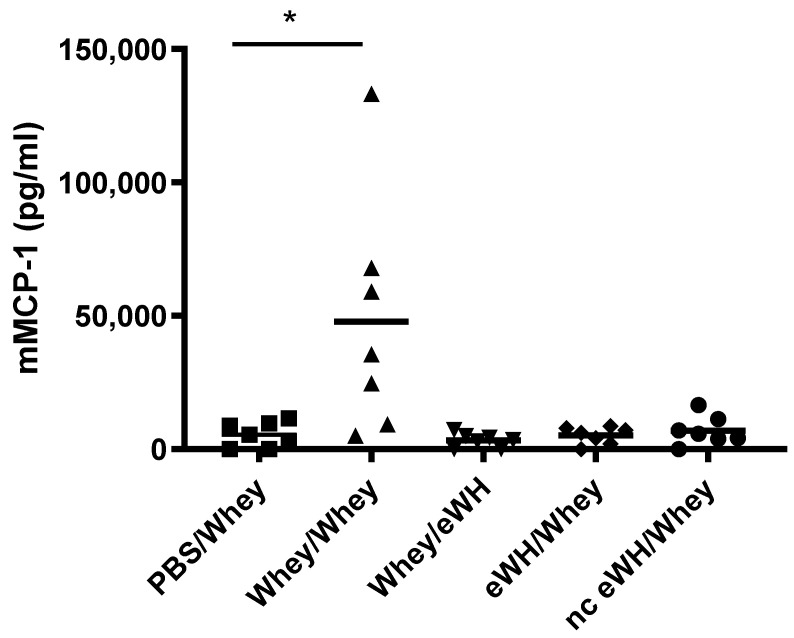
Mucosal mast cell activation by mMCP-1 level analysis in serum (day 36). Mice were orally sensitized to whey or whey hydrolysates and then challenged with intact whey or hydrolysates. After challenge sera were obtained, mMCP-1 was measured by ELISA. Data are expressed in pg/mL as individual values. ANOVA for treatment was significantly different; post hoc Bonferroni’s multiple testing showed significance * *p* < 0.05 compared to non-sensitized mice (PBS/Whey).

**Figure 7 foods-11-02005-f007:**
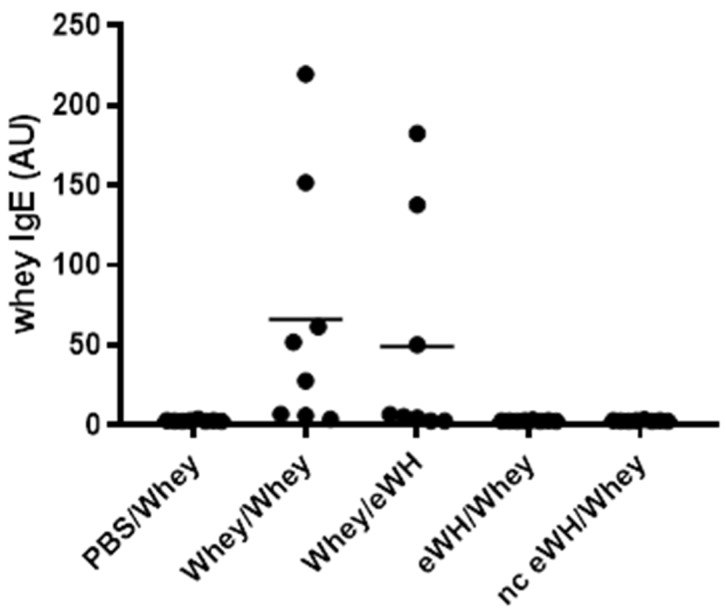
Whey-specific IgE levels in serum. Mice were orally sensitized to whey or whey hydrolysates and then challenged i.d. with intact whey or hydrolysates. Levels of whey-specific IgE were determined in sera collected on day 36. Data are expressed in arbitrary units (AU) as individual values.

**Figure 8 foods-11-02005-f008:**
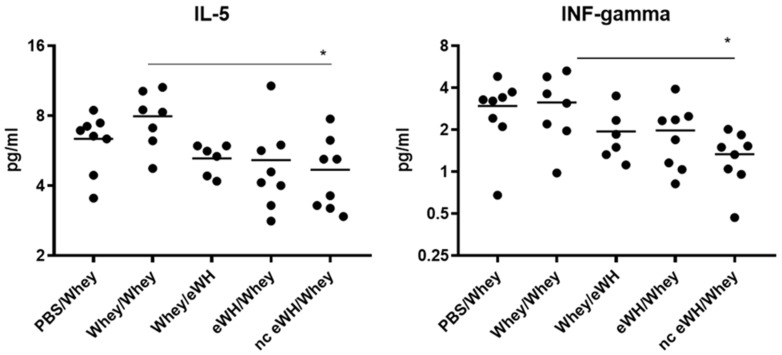
IL-5 and IFN-gamma concentrations in serum. Concentration of IL-5 and IFN-gamma were measured in serum collected on day 36 with a multiplex Th1/Th2 assay. Data are expressed in pg/mL as individual values. ANOVA for treatment was significantly different; post hoc Bonferroni’s multiple testing showed significance * *p* < 0.05 of nc eWH/whey compared to sensitized mice (whey/whey).

**Table 1 foods-11-02005-t001:** Molecular weight distribution (%) and concentration of BLG (ppm) of the extensive whey hydrolysate (eWH) and negative control eWH (nc eWH). Mean ± SD are shown.

Molecular Weight (Da) (*n* = 3)	eWH	nc eWH
>10,000	0	0
5000–10,000	0	0.2 ± 0.1
3000–5000	0.2 ± 0.1	4.4 ± 0.2
1000–3000	13.6 ± 2.3	37.5 ± 1.5
500–1000	27.2 ± 0.8	24.5 ± 1.1
<500	58.9 ± 2.9	33.4 ± 2.5
(BLG) (*n* = 2)	0.06 ± 0.007	0.09 ± 0.0007

## Data Availability

The data presented in this study are available on request from the corresponding author.

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
