# Peer review of "An In Vitro and In Vivo Translational Research Approach for the Assessment of Sensitization Capacity and Residual Allergenicity of an Extensive Whey Hydrolysate for Cow’s Milk-Allergic Infants"

_foods, 2022, doi:10.3390/foods11142005_

Round 1

Reviewer 1 Report

The purpose of this paper is difficult to fathom. It shows information similar to that used previously to assess the possible allergenicity of an undisclosed extensively hydrolysed milk preparation (by an unknown method) with a control which is is apparently another undisclosed but less extensively hydrolysed preparation. It perhaps wants a publication that can be used to endorse the reduced allergenicity of the first hydrolysate. A clear purpose is needed.

At least from the Western blotting and the size filtration it can said that preparations examined were extensively hydrolysed. This however could be better discussed by reference to the paper of Nutten et al (includes P.J. Simon as co-author) doi:10.1111/all.14098 on size versus break through allergenicity.

The assay with ELISA kit for BLG is not informative without further information on the antibody used in the kit and how the sensitive to the hydrolysis of the BLG. Presumably  the kit Beta Lactoglobulin ESMRDBLG-48" was used. Please specify.

The description of the RBL-huFcεRI degranulation assay using (apparently) six BLG lactoglobulin-specific mouse monoclonal antibodies chimerised to human IgE kappa anti- bodies should be more completely described to make it clear that they originated from a single strain of mouse and perhaps a single mouse. The discussion should also disclose how well these represent human antibodies. Ref 12 did not do inhibition tests to test this, only inferring possible similarly to human antibodies because mouse and human antibodies unsurprisingly bound some of the same peptides. The discussion should also discuss  how this result with BLG might related to other CM allergens, especially since not all children tolerate extensively hydrolysed milk. Are the chimeric humanised antibodies available from Bioceros?

The mouse experiments conducted along the lines of those by van Esch et al. DOI: 10.1111/j.1399-3038.2009.00924.x do show data for reduce allergenicity at sensitisation and challenge which is entirely expected from a wide range of literature. The differences in the cytokines are small and meaningless without a time course and follow-up experiments. With respect to the mouse IgE reactivity more than one strain would be needed to relate to humans.

The residual allergenicity found in the ewhey in the ear swelling reaction should be better discussed.

Data on mice sensitised with ewhey and challenged with ewhey would be of interest with respect to new antigenic determinants produce by the hydrolysation or residual allergenic peptides.

The clinical data now published in reference 11 showed the hydrolysate fitted the clinical definition of a suitable extensive hydrolysate but excluded children with anaphylaxis. Perhaps this could be commented on especially with respected with the purpose of the surrogate testing presented in this paper.

Author Response

  1. The purpose of this paper is difficult to fathom. It shows information similar to that used previously to assess the possible allergenicity of an undisclosed extensively hydrolysed milk preparation (by an unknown method) with a control which is apparently another undisclosed but less extensively hydrolysed preparation. It perhaps wants a publication that can be used to endorse the reduced allergenicity of the first hydrolysate. A clear purpose is needed.

We would like to thank the reviewer for taking the time to review our manuscript and raising some fair points. We will try to answer these questions to our best. The main purpose of performing this toolbox of assays is to fully characterize the new hydrolysate before it will be assessed in an animal study and thereafter in a clinical study, this direct comparison of the same hydrolysate in the different studies gives accurate information about the clinical relevance of the results and thereby predictability of (residual) allergenicity. A second purpose which might have to be more emphasized is that many assays are not suitable for a hydrolysed milk matrix due to severe matrix effects and that we now have a toolbox of well developed and validated assays specifically suitable for cow’s milk hydrolysates (a review/position paper on accurately assessing safety of CM hydrolysates will follow immediately after this manuscript).

Another reviewer also mentioned the point of the difference in hydrolysate. Unfortunately this is competitor-sensitive information and we cannot describe the production process in a manuscript. We only describe the complete process and (pre)clinical results to EFSA for approval of the claim, they will also redact this sensitive information in their published scientific position. For the reviewer to be able to assess the hydrolysates in this manuscript I can confidentially say that the new hydrolysate is made with a mixture of microbial endo- end exoproteases and the control with tryptic enzymes. Another difference is the pore size of ultrafiltration which is smaller in the new hydrolysate, this mainly contributes to the smaller peptide size as seen by molecular weight distribution. We hope the reviewer understands this sensitive information cannot be described in the manuscript. We added as a last sentence of the introduction (line 68-71): ‘With this translational strategy from in vitro to in vivo and eventually confirmation in a clinical study we demonstrate the clinical relevance of the in preclinical results which will improve predictability of (residual) allergenicity of newly developed cow’s milk hydrolysates.

  1. At least from the Western blotting and the size filtration it can said that preparations examined were extensively hydrolysed. This however could be better discussed by reference to the paper of Nutten et al (includes P.J. Simon as co-author) doi:10.1111/all.14098 on size versus break through allergenicity.

Indeed the WB and GPC shows that the hydrolysate is extensively hydrolysed and ultrafiltrated, and the ELISA shows reduced antigenicity and RBL reduced allergenicity of the hydrolysate, which are more biologically relevant assays due to binding to antibodies and not only peptide sizes.

The RBL-assay was developed using concentrations of hydrolysates ‘as consumed’ with hydrolysates with known positive or negative allergic reaction in either animal studies or CMA patients. This was confirmed in a ringtrial with hydrolysates with also known allergenicity, kindly provided by Friesland Campina and tested blinded. These in vitro results in this manuscript were thereafter confirmed in an animal study and clinical study showing clinical and translational value of the results. However, by Nutten et al. the RBL-assay was adapted and the way the assay was performed and calculated/interpreted was different than how the assay was developed and validated originally. In these studies a reference curve of intact BLG was produced and samples were calculated with a 4-PL-fit to have a result in ‘allergenicity µg/ml’. The reference curve concentration is up to 1 µg/ml, whereas samples have a concentration up to 10,000 µg/ml and makes extrapolation unreliable. In a comparison of allergenicity of different hydrolysates it is demonstrated that hydrolysates with higher BLG ELISA results show lower ‘allergenicity’ result than hydrolysates with low BLG ELISA results. This indicates that these assays seem not complementary and more validation should be performed and clinical relevance of the ‘allergenicity µg/ml’ should be assessed before the method is used to compare different hydrolysates for their allergenic capacity.

  1. The assay with ELISA kit for BLG is not informative without further information on the antibody used in the kit and how the sensitive to the hydrolysis of the BLG. Presumably the kit Beta Lactoglobulin ESMRDBLG-48" was used. Please specify.

The supplier of the BLG ELISA (ELISA systems) will not disclose information of the used antibody in the kit, however we have assessed this antibody with epitope mapping finding binding at the allergenic epitope of BLG (data cannot be shown due to confidentiality). The BLG ELISA is fully validated and the following is added to the M&M (line 75-76): ‘This BLG ELISA has been fully validated for the use of cow’s milk hydrolysates by Merieux NutriSciences with a LOQ of 0.2 ppm.’

  1. The description of the RBL-huFcεRI degranulation assay using (apparently) six BLG lactoglobulin-specific mouse monoclonal antibodies chimerised to human IgE kappa anti- bodies should be more completely described to make it clear that they originated from a single strain of mouse and perhaps a single mouse. The discussion should also disclose how well these represent human antibodies. Ref 12 did not do inhibition tests to test this, only inferring possible similarly to human antibodies because mouse and human antibodies unsurprisingly bound some of the same peptides. The discussion should also discuss how this result with BLG might related to other CM allergens, especially since not all children tolerate extensively hydrolysed milk. Are the chimeric humanised antibodies available from Bioceros?

Epitope mapping is a generally accepted methods to assess binding of antibodies to specific epitopes of a protein. With this epitope mapping was found that the chimeric antibodies bind to similar epitopes as the IgE from CMA patients bind, showing it very well represents the human allergic situation. In the publication of Knipping, PLoS One 2014 was already described: Recently, five major linear BLG epitopes/regions, covering sequences of 12–14 amino acids in length, have been identified by polyclonal serum IgEs from CMA patients (Jarvinen, Int Arch Allergy Immunol, Cocco, Clin Exp Allergy 2007). Interestingly, four of these five major allergenic’ BLG epitopes/regions were shown to be overlapping epitopes/regions found with our chimeric human IgE anti-BLG antibodies, which also underscored the allergenic relevance of our generated human IgE anti-BLG antibodies.’ In the discussion is added (line 337-340): ‘This was further substantiated by the absence of in vitro allergenicity of the eWH in the degranulation assay using RBL-huFcεRI cells which has proven human relevance since the pool of chimeric antibodies cover the allergenic epitopes as assessed with epitope mapping with IgE of CMA infants and known from literature [12].’ The chimeric antibodies are not commercially available, the execution of the complete RBL-assay is commercialized by Polpharma Biologics (previously Bioceros) and will be performed upon request.

The reviewers raises a good point of the other CM related allergy. Since this hydrolysate is whey-based, another (minor) allergen to consider is ALA. From previous experiments we know that during the hydrolysis BLG and ALA are broken down at the same rate (With SDS-PAGE and HPLC was used to analyze the chemical fingerprint of the whey hydrolysates (time points 0, 10, 15 and 30 min)). In the discussion is added (340-344): ‘It should be noted that the other allergen in whey, ALA, is only assessed with the immunoblotting when using anti-ALA antibodies, both the ELISA and RBL-huFcεRI are specifically investigating BLG. In a previous study, however, we have established that during the hydrolysis, BLG and ALA are broken down at the same rate.’

  1. The mouse experiments conducted along the lines of those by van Esch et al. DOI: 10.1111/j.1399-3038.2009.00924.x do show data for reduce allergenicity at sensitisation and challenge which is entirely expected from a wide range of literature. The differences in the cytokines are small and meaningless without a time course and follow-up experiments. With respect to the mouse IgE reactivity more than one strain would be needed to relate to humans.

We agree with the reviewer that the cytokine differences are very small and only a ‘snapshot’ but since they were measured and small differences were found we feel obliged to mention these. To the discussion is added (line 378-379): ‘The significance of these cytokines should be further explored in a time course manner in the CMA mouse model.’ We are aware that different mouse strains can show different reactions, we have for the safety assessment of hydrolysates chosen for this particular model since this is extensively tested for hydrolysate research and has also been assessed for its robustness by doing an interlaboratory evaluation (van Esch B. Interlaboratory evaluation of a cow's milk allergy mouse model to assess the allergenicity of hydrolysed cow's milk based infant formulas. Toxicoll Lett 2013 Jun 20;220(1):95-102) , for none of the other models any validation of the model was done.

  1. The residual allergenicity found in the ewhey in the ear swelling reaction should be better discussed.

A residual ear swelling was indeed found, however it does not seems to be linked to the allergic reaction since there was a complete absence of any allergic symptom or presence of IgE. It might be that other immune cells are involved in this swelling which might have been attracted to the injection site, however we have not investigated this and are therefore not able to explain in detail. The discussion has been changes somewhat to make more clear it is an immune reaction but nor contributing to the allergic reaction.

  1. Data on mice sensitised with ewhey and challenged with ewhey would be of interest with respect to new antigenic determinants produce by the hydrolysation or residual allergenic peptides.

Indeed scientifically a very interesting question, however out of scope for this manuscript

  1. The clinical data now published in reference 11 showed the hydrolysate fitted the clinical definition of a suitable extensive hydrolysate but excluded children with anaphylaxis. Perhaps this could be commented on especially with respected with the purpose of the surrogate testing presented in this paper.

Although not forbidden, the Medical Ethical Committee will not allow the inclusion of infants with previous anaphylactic reactions to cow’s milk, since there might be a possibility of a recurring anaphylactic reaction. The target group for extensively hydrolysed formula is infants with mild to moderate CMA, infants with severe CMA are only allowed in studies with amino-acid based formulas.

Reviewer 2 Report

This is a well-conducted, -analyzed and -written work.

I have no major comment to address to the authors, except the lack of indications about the process used for preparing the so-called “extensive whey hydrolysates” eWH: enzymes used? enzyme/protein ratio? T°C used? duration of hydrolysis?.... This is important to establish some correlation between the length of residual peptides remaining in the hydrolyzates and the putative enzymatic cleavage sites occurring along the aa sequences of BLG and ABL. This point should be discussed in the Discussion section of the paper.

Author Response

We would like to thank the reviewer for taking the time to review our manuscript. The reviewer touches upon an important but very difficult point which is the enzymes used and hydrolysis process as we are very aware that this is important information. However, this information is very competitor-sensitive information which we will not disclose in a publication. We only describe our complete process and (pre)clinical results to the EFSA for approval of the claims, they will redact also this sensitive information in their published scientific opinion. For the reviewer to properly assess the content of our manuscript I would like to give confidentially some more information, the new hydrolysate is hydrolysed with a mixture of microbial endo- and exo-protease and the control hydrolysate is from tryptic hydrolysis. Another important difference is the pore size of the ultrafiltration which is smaller in the new hydrolysate. The latter will be the most important for the difference in molecular weight/peptide sizes. We hope the reviewer is content with additional information without having to add this information to the manuscript.

Reviewer 3 Report

Foods

 Article 

An in vitro and in vivo translational research approach for the assessment of sensitization capacity and residual allergenicity  of an extensive whey hydrolysate for cow’s milk allergic in fants

The authors studied the management of cow’s milk allergy (CMA) in children. Both in-vitro assays and an in-vivo mouse model for CMA were used to assess the sensitizing and allergenic potential of a newly developed extensive whey hydrolysate (eWH).

This was subsequently confirmed in a clinical study in which this eWH was tolerated by more than 90% (with 95% confidence) of infants or children with confirmed CMA.

English and grammar are very good.

No mistakes found

An adequate number of tables and figures were presented.

Accept it as it is, with a single correction in fants is a single word.

Author Response

We would like to thank the reviewer for taking the time to review our manuscript and we are happy to hear it is to the reviewer's satisfaction. Infants is indeed a single word, I noticed a sort of auto-break and we have noticed the journal about this.